# The Influence of Visibility on the Opportunity to Perform Flight Operations with Various Categories of the Instrument Landing System

**DOI:** 10.3390/s23187953

**Published:** 2023-09-18

**Authors:** Anna Kwasiborska, Mateusz Grabowski, Alena Novák Sedláčková, Andrej Novák

**Affiliations:** 1Faculty of Transport, Warsaw University of Technology, 00-662 Warsaw, Poland; 2Faculty of Operation and Economics of Transport and Communications, Air Transport Department, University of Žilina, 010-26 Žilina, Slovakia

**Keywords:** meteorological conditions, navigation system, ILS, atmospheric visibility, airport, airline, diverted flight, GNSS, weather, aviation, METAR data

## Abstract

Meteorological conditions significantly affect air traffic safety and can also affect a pre-planned flight plan. Difficult meteorological conditions are particularly hazardous during take-off and landing procedures. Still, they can also cause disruptions to air traffic by causing, for example, delays to air traffic or diversion of aircraft to other airports. From the airlines’ point of view, such situations are not beneficial if flights are diverted to other airports due to reduced visibility at the airport caused by fog and haze. For flight operations, a popular navigation system with a precision approach is the ILS, which has several categories enabling an approach even in adverse meteorological conditions. However, not every airport has a high-category ILS, and setting up such navigation equipment is lengthy and costly. The main objective of this article is to analyze the impact of meteorological conditions, particularly visibility, on the possibility of performing flight procedures with different ILS categories. The study was designed to quantify the limitations associated with meteorological conditions with specific ILS equipment at a given airport. The research questions for this study include the following: What were the meteorological conditions in terms of visibility? What impact did the visibility parameter have on the performance of landing operations at the airport under study? Can an indication of the probability of stopping landing operations be important in recommendations for scheduling airline flights to avoid delays? Three airports were selected for the analysis: Warsaw Chopin Airport, Warsaw Modlin Mazowiecki Airport, and Krakow John Paul II Airport. The analysis was based on approximately 52,000 METAR dispatches in 2019 and 2022. The research indicated during which periods landing procedures were most frequently halted and calculated such a change with a different category of ILS. For the Kraków Airport, the probability of stopping landing procedures in any month was calculated, along with recommendations for flight schedule planning for this airport. The research results can be used to better plan airline flight schedules, avoiding hours with a high probability of reduced visibility, which may result in rerouting flights to another airport. Long-term low clouds and reduced visibility affect the safety of operations but also cause delays.

## 1. Introduction

Aviation is well known to be highly dependent on the weather and temporary weather conditions. The current meteorological situation is primarily important during take-off and landing procedures [1,2,3]. The meteorological information and conditions influence air traffic management, the planning of airlines and airports’ operations, and the punctuality and safety of flight operations. How meteorological information is prepared and presented is standardized, as such information must be readable by all air traffic users. This information is provided in the form of dispatches, influences flight planning, and is used by flight crews before take-off and landing and during the flight [4]. Some atmospheric phenomena can significantly complicate and often even prevent flight operation. In addition, they pose a risk to both the aircraft and flight crew and passengers on board. The on-board meteorological equipment that has been put in place cannot always handle all weather-related events, such as high winds, turbulence, and visibility [5]. Meteorological conditions are important for the performance of airport flight operations and can affect flight procedure disruption [6]. To a large extent, they determine the operational capabilities of air transport and, in a long-term analysis, determine the safety of flight operation and procedures (at each stage from take-off, through in-flight conditions, to landing) and the magnitude and frequency of air traffic delays, which have obvious economic and social significance [7]. Adequate navigational equipment at airports and providing aircraft with on-board equipment that cooperates with ground-based equipment make it possible to cope with and carry out flight operations in different and often difficult meteorological conditions. To clearly define the conditions under which such operations can take place, minimum weather conditions under which a take-off or landing can be performed are set for each aerodrome. Respecting and adapting to the prevailing weather conditions and the specified minima increases the safety of the execution of operations. Despite these safeguards, bad weather conditions should not be underestimated, as there have been aviation accidents in the history of aviation where the cause (or a very important premise) was attributed to bad meteorological conditions. It should also be noted here that the causes of an aviation accident are often diverse and do not depend on just one factor, including meteorological. We followed cases for our research where one of the factors causing the accident was meteorological.

The paper chose visibility as an essential aspect that can impact air traffic. The main objective of this paper is to present a quantities analysis of the influence of visibility on the opportunity to perform flight operations with various categories of the instrument landing system. The results of the research analysis are an indicator of the probability of stopping landing operations, which can be important in recommendations for scheduling airline flights to avoid delays. 

The paper is organized as follows. Section 1 refers to the introduction. Section 2 contains a literature review. Section 3 provides an overview of the impact of meteorological conditions on air traffic performance. Section 4 presents the sources of data and a description of the methodology. Section 5 contains the results of assessing the impact of the ILS category on the performance of landing at selected airports with input data. Finally, the last sections present the conclusions and the proposed directions for future work.

## 2. Literature Review 

### 2.1. Legal Background of Aircraft Accident and Incident 

For a better understanding of the theory, it is necessary to describe the definitions of accident and incident as defined in Annex 13 published by the International Civil Aviation Organization (later as “ICAO”). ICAO is an organization developing the standards and recommendations for civil aviation named Annex No. 1–19, which are the amendments to the Convention on International Civil Aviation from 1944. This convention is to date the most important international legal document for civil aviation. 

Annex 13 of ICAO focuses on aircraft accident and incident investigation and defines aviation accidents and incidents [8]. An accident is an event related to the operation of an aircraft in which a person is fatally or seriously injured, in which the aircraft sustains damage or structural failure requiring repair, and when the aircraft is missing or completely inaccessible. An incident is defined in Annex 13 as “An occurrence, other than an accident, associated with the operation of an aircraft which affects or could affect the safety of operation”. A serious incident is defined as “An incident involving circumstances indicating that there was a high probability of an accident” [9].

The other important document is ICAO Annex 3, which describes the meaning of and general provisions for meteorological service and the provision, use, quality management, and interpretation of meteorological information, which are highly important. 

The literature contains many research results on aviation events and the impact of meteorological conditions on aviation accidents and incidents. One of them concerns research related to in which phase of aircraft flight the most aviation incidents occur. The background of the research [10] is based on the results of conducted research between 2009 and 2018, which showed that landing and approach operations in general aviation are at risk of accidents (Figure 1). 

So, for the current research, it was interesting to focus on the analysis of data from the Polish National Transportation Safety Board (later as “NTSB”), which investigated weather-related aviation accidents in various phases of flight based on 17,325 reviewed accidents and incidents and the Aviation Safety Reporting System (later termed “ASRS”). The study aimed to quantify and describe weather-related accidents and incident reports involving on-board meteorological equipment-equipped aircraft operating under Part 91 General Aviation Flight rules using publicly available government accident reporting sites. There were 1382 weather-related accidents and incidents throughout this investigation in different phases of flight. The phases of flight with the highest deaths were maneuvering and enroute and then approach and take-off. Further results were published by [11] in 2023, where it was declared that 6.12% of all accidents in the years 2016–2021 in Czechia and Slovakia were caused by meteorological phenomena. It is not only dangerous for flight safety but also for future air transport operation and economics when growing weather-related enroute delays influences them [12]. 

It should be noted that not only weather conditions are the cause of aircraft accidents and incidents. Nevertheless, there is a need for effective meteorological forecasting and meteorological services. It should be noted that there are also other causes of aviation accidents and incidents, which include, for example, an unstable approach to landing, technical problems with the aircraft, or human errors on the part of pilots or air traffic controllers. Such factors may also contribute to a missed approach and diversion of the aircraft to another airport.

### 2.2. Aviation Incidents and Accidents Related to Weather Conditions

In the United States, weather is responsible for 87% of ground delays, of which 23% are due to adverse wind conditions, 35% to low cloud bases, 16% to thunderstorms, 8% to snow and icing, 7% to limited visibility, and 11% to other factors [13]. The low height of clouds causes limited visibility and affects the performance of air operations [14,15]. Limited visibility at low levels causes disruption during landing and take-off, and in some situations can bring air traffic to a complete standstill. Taking the example of San Francisco International Airport, with good visual conditions, up to 60 arrivals per hour are possible. However, with low clouds (stratus and fog), the efficiency drops to 25–30 arrivals per hour [16]. When the cloud ceiling or visibility is very low, an arriving aircraft may not be able to land (and diverted to an alternate), and a departing aircraft may not take off until conditions improve [17]. Studies of low cloud base formation in Europe [18,19] have shown that low cloud bases are most common in north-western, central, northern, and eastern Europe and less common in the Mediterranean area. Unfavorable meteorological conditions may be the cause of aviation accidents. The research conducted in [20] was aimed at identifying the causes, causative factors, and related problems of weather-related accidents in general aviation (GA). The results suggest that the most common factors causing fatal weather accidents are low cloud ceiling (20%), fog (14%), wind (10%), and night (9%).

The data show that bad meteorological conditions caused 40% of aviation accidents. Of these, 40% were 66% caused by low visibility and a low cloud base [21]. Evidence that low visibility poses a particularly high risk is provided by the 1969 air accident of PLL flight LOT149, in which the pilot of the aircraft ignored information about visibility, which had dropped to 400 m. The permissible minimum visibility at the time was 1100 m. As a result, the aircraft’s right wing hit a tree, and the whole aircraft tilted to the right at a 45-degree angle [22]. No one was killed in the aircraft accident, but the damaged aircraft was disabled for further operation. The cause of the accident was the approach to the landing below the applicable visibility minima.

In 1977 [23], the most tragic accident in aviation history occurred in the Canary Islands, killing more than 550 people. Tenerife Airport was very busy on the day of the accident due to the diversion of aircraft following a bomb alert on a neighboring island. The airport was not prepared to handle such heavy traffic, and, in addition, there was dense fog in the area, which significantly reduced visibility. In these difficult conditions, two Boeing 747s—of Dutch airline KLM and American PanAm—collided. However, the collision occurred not in the air but on the runway. The most significant causes of this accident were declared to be human factors and communication problems between air traffic controllers and pilots, but dense fog also severely limited visibility.

In 1997 [24], an aircraft accident occurred on Garuda Indonesia flight 152. In the second half of that year, Southeast Asia was hit by massive smog clouds caused by forest fires, which severely reduced visibility in Indonesia. The smog peaked in late September and early October. In these conditions on 26 September, an Airbus A300 was approaching to land at Medan Airport. During the investigation, it was indicated that the cause was massive smog clouds limiting visibility for aircraft, which means that the cause of the accident was the reduced visibility and not the meteorological conditions themselves.

In 2003, an Il-76 crashed in mountainous terrain near Kerman, Iran. A strong wind was reported in the area where this aircraft was flying, and the aircraft then disappeared from radar screens. At that time, residents in the area heard a loud explosion. The investigation indicated that the cause of the accident was difficult weather conditions, particularly very strong wind blowing along the flight path [25].

Difficult meteorological conditions can create safety risks but also flight delays due to diversion to other airports. Challenging conditions occurred in December 2022 where aircraft were unable to land at Gdansk Airport. In the case of Gdansk Airport, the RWY29 runway is equipped with a category IIIB ILS. The RWY11 runway has less precise landing approach procedures based on GNSS and VOR/DME. Landing minima were insufficient for these systems. Some flights were diverted to other airports. Some aircraft had to perform holding and perform a go-around. The situation caused delays of several hours [26].

Aircraft incident no. 962/09 occurring at Katowice Pyrzowice Airport is an excellent example of correctly reading meteorological information and adapting to the prevailing meteorological conditions. In the evening hours in difficult atmospheric conditions in 2009, according to the METAR dispatch (at 19:30), there was fog at the airport, and visibility along the runway from the RWY27 threshold was 350 m. Therefore, the ILS cat. II landing approach conditions were not approved at the airport. At that time, a Boeing 737–800 crew flying from Fuerteventura was performing a cat. I ILS approach to RWY27. The autopilot was disengaged at 89 ft (27 m) above airport level, while the automatic engine power control was disengaged at 33 ft (10 m). At an altitude of 50 ft (15 m) above the runway threshold, the pilot flying started to turn left, then at an altitude of 30 ft (9 m), started to return to the centerline by making a right turn (following the first officer’s command—fly right). After touchdown on the left side of the runway center line, the aircraft with the left main landing gear and the nose landing gear partially left the runway over a length of 373 m. During this time, the wheels of the right main landing gear remained on the asphalt edge of the runway at all times. The aircraft then returned to the runway and completed its landing run [27].

The State Commission on Aircraft Accidents Investigation determined that the cause of the accident was that the crew continued the precision approach below the applicable decision height (DH) in conditions where the RVR was 375 m and below the published and required value of 550 m. As a result of this accident, the aircraft was rendered inoperative, damaging the main and front landing gear and the lighting parts of the airfield navigation aids. This incident clearly demonstrates how visibility and appropriate pilot behavior affect the safe execution of flight operations. Visibility can be reduced due to the presence of fog, haze, or turbidity, making it difficult to perform take-off or landing procedures. The primary reason for reduced visibility is condensation, leading to fog, haze, and clouds.

The chosen air accidents are only examples of incidents where meteorological conditions had a significant impact. However, it is important to note that bad meteorological conditions were not considered to be the main cause of these accidents. The literature review contains many statistics on the causes of accidents, which cannot be listed in full. Determining whether the cause was direct or only contributory is also important. In ongoing air accident investigations, each case is analyzed individually. It should be noted that the existence of several independent root causes can lead to an aircraft accident.

## 3. The Impact of Meteorological Conditions on Air Traffic Performance

### 3.1. Atmospheric Phenomena

Weather forecasting is therefore essential for the operation of airports and the operation of aircraft to the airport or the diversion of flights to other airports if the meteorological conditions do not ensure a safe landing [28]. Meteorology and its conditions are therefore becoming an important part of flight planning [29], and the implementation of suitable equipment and systems in the aircraft can increase the safety of flights based on more accurate information about weather conditions [30,31]. Throughout the year, there can be a wide variety of meteorological conditions, which can more or less significantly affect the occurrence of days with reduced visibility due to low cloud bases or the presence of so-called hydrometeors in the air, i.e., atmospheric phenomena related to the presence of water in the atmosphere. These phenomena form part of a broad package of meteorological data and information that is regularly measured and observed to provide airspace users with aviation meteorological cover [32]. They are provided in the form of so-called aeronautical–meteorological documentation for pre-flight planning for use by flight crew members prior to take-off and by aircraft in flight. Precision and a detailed weather forecast are therefore essential for the safety and efficient operation of air carriers and airports. Particularly important atmospheric phenomena for landing procedures are [33,34] wind, visibility, icing, turbulence, temperature and dew point temperature, air density, precipitation, and other phenomena such as thunderstorms. For the purposes of this research, the visibility parameter is included because it is a significant parameter determining the minimum conditions for flight operation. Visibility at airports is very important for pilots, who need a visual reference to complete the landing [35,36]. Visibility is a measurable characteristic provided by an observer or automated system, expressed in the aeronautical–meteorological documentation as a four-digit notation in meters or kilometers, which defines the longest distance from which an object with certain characteristics is visible and recognizable [37]. If horizontal visibility measuring systems are used, visibility should be measured at a height of approximately 2.5 m above the runway. Visibility measurement sensors for local regular and special announcements are located to best indicate the visibility representative of the runway and the touchdown zone. Visibility information is communicated by the airport tower air traffic controller via radio to the pilot, both during take-off and on approach. 

### 3.2. Type of Visibility

There are different types of visibility in meteorology. It is the distance at which it is possible to perceive and identify the given object depending on the object of observation, the state of the air, and the physiological conditions of the observer. The visibility of objects against the ground surface and objects in the air is called aerial visibility and (Figure 2) is divided into horizontal, vertical, and slanted visibility.

Horizontal visibility is the distance at which an observed dark object near the horizon is still visible and recognizable against the sky. In connection with flights in difficult atmospheric conditions, the concept of landing visibility has been introduced, understood as the limiting distance defined along the landing path from which the pilot on board the aircraft can recognize the runway, determine its direction, and switch from instrument flying to piloting with visibility. Vertical visibility (VV) and oblique visibility are the limiting distances from which a viable object on the ground can be seen from the aircraft.

An additional runway visibility parameter is the RVR (runway visual range), which is defined as the distance from which the pilot of an aircraft on the runway centerline can see the runway markings and the lights outlining the runway or its centerline. The instrument systems used to determine RVR are called visibility meters. RVR is measured when the visibility is less than 1.5 km. The traditional way of determining or estimating RVR is by counting visible marker lights along the runway. According to the regulations, the RVR visibility test is performed one hour before the aircraft is due to take off and a few minutes before landing. Naturally, for the landing procedure, the aircraft must have the minimum operating conditions for landing, i.e., it must have on-board equipment for such landing. It must be considered that the category of the aerodrome and the corresponding equipment with navigational aids determine the necessary minimums and the pilots’ decisions.

Visibility is considered one of the most important meteorological phenomena. In cases of low vertical visibility and RVR, the pilot of an aircraft may miss characteristic objects on the descent path. This translates into incorrect execution of necessary approach operations, e.g., course correction, circle building, or exit to the descent path. In spite of advanced navigation systems (ILS), low visibility can affect the late sighting of the runway threshold, making it impossible to perform the landing. For performing a landing in limited visibility, the decision height is important in determining whether a landing is possible. Low visibility strongly impedes or completely prevents landing (depending on the navigation system category).

According to the current regulations, the basic condition for low visibility procedure (later termed “LVP”) operations is implementing and applying appropriate low visibility procedures at the airport [38]. These procedures apply to approach operations in atmospheric conditions below category I standard, outside category II standards, in category II and III standards, and take-off when the runway visual range (RVR) is less than 400 m but not less than 75 m. Performing flight operations in such conditions poses an additional risk to the safety of flight operations. The pilot of the aircraft during approach has a very reduced time to visually assess the airport conditions and make a decision for landing. The performance of LVP operations requires the aerodrome operator to adapt the aerodrome infrastructure in advance and to meet several technical requirements concerning, inter alia, the physical characteristics of runways and taxiways, visual aids, electrical power systems including emergency power supply, and aerodrome obstacle containment surfaces. An aerodrome operator may not authorize LVP operations if the aerodrome is not adequately prepared for this, i.e., does not have adequate services, facilities, equipment, and authority-approved and implemented LVP procedures at the aerodrome. An LVP procedure may be developed for an aerodrome equipped with ILS cat. II and III. The procedure may be implemented if the visibility along the runway drops below 550 m and/or the cloud base covering more than half the sky drops to below 200 ft (60 m). In that case, the LVP introduction and the need to switch on the appropriate runway, taxiway, and stop bar lighting must be communicated.

### 3.3. Visibility Limitations Depending on the Phenomena

The main causes of visibility restrictions include the occurrence of fog, haze, precipitation, winds lifting sand and dust particles, and a low cloud base, as in Table 1.

Fog is a community, a suspension of condensation products in the ground-level layer of the atmosphere, the concentration of which limits visibility in a horizontal direction below 1 km. Fog is formed when conditions are created near the ground surface that favors the condensation of water vapor. Conditions favoring the formation of fog are a small difference between the temperature of the air and its dew point temperature (below 4 °C), weak surface wind, cold surface air, and humid (warmer) air above it.

Haze is a phenomenon of limited visibility involving byproducts of condensation between 1 km and 10 km, both close to the ground and at a certain height in space. Heavy haze near the ground is dangerous because the pilot, initially seeing the aerodrome well from a certain height, does not anticipate a deterioration in visibility. Meanwhile, as it descends, visibility rapidly decreases, and at some point, the pilot may lose visibility of the runway, making landing impossible. 

Fog and haze may occur more frequently in areas near rivers and lakes but can form over much larger areas affecting airport operations. Such phenomena affect all airports, even those located at a distance of 55 km from Warsaw/Modlin Airport, which has also repeatedly encountered problems with the non-acceptance of aircraft due to lingering fog. For example, in October 2012, fog paralyzed the operation of many Polish airports for many hours. Due to bad weather conditions, airports in Bydgoszcz, Poznań, Wrocław, Warsaw, and Modlin did not accept aircraft, and Chopin Airport was in normal operation only in the afternoon. Fog also caused many disruptions at airports close to each other, i.e., Warsaw Chopin Airport and Warsaw/Modlin Airport. On many occasions, despite the installation of high-end navigation equipment, even Chopin Airport was unable to handle and accommodate aircraft at the airport due to very dense fog. Gdańsk Airport has also had to deal with restricted air traffic due to dense fog, even though this airport is equipped with a state-of-the-art navigation system. Fog is the worst bane of airports, as visibility is drastically reduced during this phenomenon. Fog is not only a problem at Warsaw/Modlin Airport but, as described above, at many airports in Poland, Europe, and the world.

Turbidity caused by airborne dust, sand, smoke, or exhaust particles entering the atmosphere under strong turbulence can also be a phenomenon that degrades ground-level visibility. Dry turbidity can reduce visibility to 4–6 km or less and can reach up to a 10–12 km altitude. With dry turbidity, relative humidity does not exceed 70%.

Visibility can be limited by the height of the clouds’ base, which does not necessarily indicate the limit of vertical visibility but is important for estimating visibility. In contrast, vertical visibility is assumed to be equal to the height of the cloud base. This parameter is, in turn, measured using equipment located on the approaches at some distance from each runway threshold on the extension of the runway axis, which prevents erroneous readings caused by measuring aircraft passing over them.

Clouds can vary in structure, and knowledge of the phenomena occurring in the various types of clouds can often help avoid the dangers and hazards associated with planned or accidental flight in clouds. Therefore, before a flight, the pilot should familiarize himself with information on cloud cover, the type of clouds on the route and in the flight area, the height of the base of each type of cloud, visibility in the clouds, and the occurrence of storm phenomena. The type of cloud is also important for aviation because, on this basis, it is possible to give the average parameters, i.e., the height of the cloud base, visibility, and the possibility of precipitation, turbulence, icing, or storm phenomena. Low-level cluster clouds (usually occurring between 30 m and 2.5 km) are stratus, cumulus, and cumulonimbus clouds (occurring above the low level and able to reach up to 12–14 km). Visibility in stratus clouds can be moderate to poor (30–100 m), and the average cloud base is around 100–300 m. Drizzle, snow, or freezing drizzle may fall from these clouds. Visibility in cumulus clouds should be classified as poor and poor—less than 10 m. Transient rain, rain with snow, and snow may occur from cumulus clouds depending on the season. Average cloud bases are 600–1000 m high, but the tops of these clouds can reach up to 4000–7500 m. The most dangerous clouds are the cumulonimbus clouds. Visibility in these clouds is very limited (less than 10 m). Lightning, very strong turbulence, and extreme icing occur in these clouds. These clouds produce intense rain, rain with snow, and sometimes hail, which can reduce aerial visibility to tens of meters. Therefore, knowing the cloud cover and the clouds present is important for estimating the feasibility of flights.

Humidity is a parameter describing the amount of water vapor in the air. Air is saturated when it has 100% relative humidity. In this respect, it is also necessary to determine the temperature. Changes in temperature at the earth’s surface cause air to move vertically (this leads to clouds) and horizontally (this leads to wind). 

The dew point temperature is a measure of the moisture content of the air. By adopting mathematical relationships from the available information on temperature and dew point temperature, it is possible to determine the approximate relative humidity and the approximate height at which the condensation process leading to cloud formation will begin. The greater the discrepancy between the outside and dew point temperatures, the drier the air. A smaller difference between these values indicates expected fog, haze, or low clouds. In contrast, other factors, such as wind, can cause fog to not linger for longer. Dew point information is of great importance to pilots as it can be used to determine, for example, the likelihood of icing, fog formation, or cloud cover limits. When the air temperature reaches the dew point level, then the formation of fog is certain.

The relative humidity of the air *f* is the quantity that defines the current vapor pressure to the maximum vapor pressure at the same temperature and constant air pressure, expressed as a percentage. Air is saturated when it has 100% relative humidity, whereas when it is less than 100%, it is referred to as unsaturated. Relative humidity can be estimated by knowing the temperature *T* and the dew point temperature *T_pr_*. 

The process of condensation on the runway surface starts when the temperature of the runway falls below the dew point temperature. Therefore, the indication of the dew point temperature is important in the meteorological dispatches available to aviation. The dew point temperature determines the air temperature value to which the air must cool to reach a state of water vapor saturation. When the air temperature reaches the dew point temperature, condensation occurs, and fog, mist, and water deposits form on the ground, vegetation, objects, vehicles, etc. (dew, hoarfrost, drifts, rime, frost, etc.). The level of condensation *P_k_* (condensation level), i.e., the appearance of cloudiness and fog, can be approximated by knowing the value of the temperature and the dew point temperature. 

As mentioned earlier, an important issue when discussing cloud cover is the cloud base. The height of the cloud base may differ from that declared by the synoptic station and from that given by the pilot. This difference is due to the reference point but also to the different densities of water droplets or ice crystals building up in the lower part of the cloud. The first possibility for error is the unevenness of the cloud base because, under natural conditions, the cloud base is not a condensation line but is undulating. The magnitude of the unevenness of the base depends, among other things, on the intensity of turbulent movements in the lower part of the clouds. The second possibility of a misreading is due to the pilot’s different perception of the cloud base. The pilot takes the cloud base, the level of total loss of ground visibility. Despite these observations, the officially reported cloud base is the basis for decision making. Cloud base and cloud formation are also important from the point of view of air traffic planning and organization. A necessary condition for the development of clouds is the unstable equilibrium of the air. This causes upward vertical air currents arising in overheated areas to carry water vapor to the condensation level. The ascending moist air at this height reaches its dew point temperature. Further cooling leads to the condensation of water vapor, which results in the development of a cloud with a very large vertical extent. An approximate estimate of the height of the condensation level for clump clouds can be made:h = 123 × (*T* − *T_pr_*) [m],(1)

Cloud base height should be understood as the distance between the ground surface and the lower limit of the cloud range. A low cloud base makes flight, landing approach, and landing more difficult. There is a view in circulation that flying in clouds is akin to moving through dense fog. Specific physical processes form each type of cloud. It is important to obtain information on cloud cover related to the amount and type of cloud (existence of turbulence and icing), the height of the base of each type of cloud, temperature, precipitation, and other information related to visibility limitations to ensure flight safety. As mentioned earlier, the most dangerous clouds for aviation are Cumulonimbus Cb clouds. They experience lightning, severe turbulence, and icing. These clouds produce heavy rain, rain with snow, sleet, and sometimes hail. During precipitation, low clouds form under the Cb clouds’ base, limiting visibility to several tens of meters. Cb precipitation clouds are the meteorological objects posing the greatest danger to aircraft in space.

Cloudiness (a parameter found in aeronautical dispatches) is defined in terms of magnitude, i.e., a measure of sky coverage. In international exchanges, cloud cover ranges expressed in octants, i.e., eighths of the sky, are used.

A low cloud base can result in precipitation, reduced visibility, and icing; hence, the occurrence of certain types of precipitation or precipitation of significant intensity is associated with a danger to flight operations. Precipitation itself is not a hazardous phenomenon for aviation. Precipitation can also cause fog (frontal fog) when relatively warm rain or drizzle falls through a layer of cool air and the evaporation of the precipitation saturates the surrounding air. Such fog can be very dense and cover large areas persisting for a relatively long time.

### 3.4. Technical Equipment

Instrument flight rules (IFRs) allow flying through clouds and other visibility constraints, but not every airport, aircraft, and flight crew can use these procedures. It is possible to land in poor visibility conditions using the instrument landing system (ILS). The ILS allows pilots to perform an instrument approach to land when it is impossible to establish visual contact with the runway. After determining the direction of the landing approach, the pilot follows the approach path given by the ILS. They then descends down the glide path to decision altitude. This is the altitude at which the pilot must have an adequate visual reference to the landing environment to decide to land or attempt to divert to another airport. They are introduced to determine the path to the target and the aircraft’s position to ensure the safety of landing procedures in limited visibility conditions and various navigation systems. Navigation systems also have a direct impact on the smoothness of landing. The most important meteorological element from the point of view of the ILS is the visibility along the runway, known as RVR, and vertical visibility (decision height). If the value of one of these visibilities falls below the permissible value (depending on the category of the ILS), landing on the runway is restricted or stopped [40].

The Instrument Landing System (later termed “ILS”) is defined as a precision runway approach aid based on two radio beams, which, together, provide pilots with both vertical and horizontal guidance during an approach to land. The ILS is a common and one of the most popular radio navigation systems today for landing in the most difficult meteorological conditions [41]. It is a radio navigation system that supports landing procedures useful in limited visibility conditions [42]. Its operation involves guiding the aircraft from the range limit to a certain point on the glide or touchdown path. The ILS categories determine the decision height expressed in meters and the RVR visibility range. The instrument systems used to determine RVR are called visibility meters. The runway visual range (later termed “RVR”) is a measurement of the horizontal visibility along the runway [43]. There are ILS categories and values of decision height (DH) and runway visibility for each system (Figure 3):I for a decision height of 60 m (system minima) and RVR visibility range of 550 m;II for a decision height of 30 m and RVR visibility range of 300 m;IIIA for a decision height of 15 m and RVR visibility range of 200 m;IIIB for a decision height of 0 m and RVR visibility range of 75 m.

The ILS is not a necessary device at the airport. It is a supporting system, but infrastructure is needed to run it, e.g., lighting.

The global navigation satellite system (later termed “GNSS”) is crucial in improving aircraft landing procedures, specifically using precision approaches. The most well-known GNSS system is the global positioning system (later termed “GPS”), but there are others like the European Union’s Galileo, Russia’s GLONASS, and China’s BeiDou [44]. The GNSS can complement or even replace the traditional ILS in some cases. GNSS can offer an additional or alternative means of precise guidance to the runway, improving accessibility to more airports and reducing reliance on ILS.

It is important to note that while GNSS systems significantly enhance aircraft landing procedures, they are not standalone solutions. They are integrated into the broader air traffic management system, including ground infrastructure and on-board avionics, to ensure safe and efficient operations during approach and landing. Additionally, pilots undergo specialized training to utilize these technologies effectively and handle any potential contingencies during critical phases of flight.

A global navigation satellite system (GNSS) can also be used in another kind of transport, for example, for positioning passenger cars, particularly in urban driving environments [45]. A GNSS can be connected with a UAV and can be an efficient tool that facilitates field measurements [46]. Using UAV is connected with predicting the UAV flight accuracy using the output graphs and monitoring the main flight phases (take-off, enroute, and landing) [47].

## 4. Material and Methods

In accordance with current regulations and practices, meteorological conditions and forecasts must be continuously monitored by services properly equipped for this purpose. Up-to-date meteorological information is essential for the smooth operation of the airport in the event of changing weather conditions. Meteorological information for flight planning is developed in the form of weather forecasts using the global area forecast system [48]. Various dispatches are developed and transmitted for the flight crews and for airport services. Participation in the exchange between airports is based on the following forecasts:Forecast for landing: TREND in code form. This occurs together with the current weather conditions information METAR with a validity of 2 h;Aerodrome forecast: TAF with a specific validity indicated in the message;SIGMET information: developed when significant weather phenomena such as thunderstorms, severe turbulence, severe icing, and volcanic clouds occur or are forecast to occur;AIRMET information: issued for low-level flights up to FL100 in the case of the occurrence or forecast of hazardous phenomena such as ground-level winds with an average speed above 60 km/h, visibility below 5000 m, thunderstorms, or low cloud base;Aerodrome warnings: in the case of hazardous phenomena;Wind shear warnings;Forecasts for flight crews: the forecast package includes maps of significant weather phenomena and sets of dispatches METAR/TREND, TAF, SIGMET, and AIRMET.

An important quantity reported in the dispatch is the horizontal visibility, given digitally in meters, and the RVR, which appears when the visibility drops below 1500 m. According to ICAO Annex 3, airports should issue METAR dispatches every 0.5 h. One of the most common reasons for not being able to carry out landing operations on a given runway is that the RVR, or decision visibility, is too low, which can be read from the METAR dispatches.

A METAR message consists of groups containing meteorological data, including visibility. The group for horizontal visibility is given in meters. For visibility between 5 and 10 km, the value is given every 1 km, while for visibility between 800 m and 5 km, it is given every 100 m. For visibility decreases below 800 m, it is given in 50 m increments, while for visibilities below 50 m, the indicator ‘0000’ is used. The group for directional changes in visibility is only given when the visibility value differs from the prevailing visibility, and the visibility is different in different directions. In such situations, the most operationally relevant direction is added at the end, e.g., 1300NE—prevailing visibility 1300 m in a north-easterly direction.

Below is an example of the METAR message structure:

LFPO 271450Z 24005 KT 170V260 CAVOK 19/02 Q1013 NOSIG RMK RWY17L 23006 KT RWY05 24005 KT 210V270 RWY23 24007 KT

The message consists of codes and abbreviations as follows: a 4-character ICAO airport identifier; day: 27; time: 14:50 UTC; wind direction: 240, speed: 05 KT, direction is variable between 170 and 260; ceiling and visibility OK, no cloud below 5000 ft (1500 m) or the highest minimum sector altitude and no cumulonimbus or towering cumulus at any level, a visibility of 10 km (6 mi) or more, and no significant weather change; temperature: 19 °C, dewpoint: 2 °C; air pressure is 1013 hPa; no significant change is expected to the reported conditions within the next 2 h; remarks; runway 17 L winds: wind direction: 230, speed: 6 KT; runway 05 winds, wind direction: 240, speed: 5 KT, wind direction varies between 210 and 270; runway 23 winds, wind direction: 240, speed: 7 KT.

The METAR dispatches were downloaded from the Iowa State University website [49]. The data obtained from METAR messages constituted the basis for the analysis. We selected the visibility parameter from the messages, which, in our opinion, is one of the determinants allowing flight operations. Visibility data were obtained from the Meteorological Aerodrome Report (later termed “METAR”) also known in the UK and USA as aviation weather report dispatches for the airports. It is a coded weather report message used in aviation meteorology and weather forecasting for containing current weather data at the airport created every half hour to provide up-to-date meteorological information on temperature, pressure, dew point temperature, wind strength and direction, precipitation, cloud cover, cloud base height, and visibility, which may also contain other information (e.g., runway conditions) [50].

This information was filtered, and the visibility and RVR data were left for further analysis. The input data were then processed and used to determine the decision height for landing operations. Information regarding the possibility of landing was linked to the minimum visibility for each ILS category. The quantitative analysis performed indicated the number of hours in which there are restrictions on the execution of landing operations for specific ILS equipment at the airport. In addition, an indicator was presented to determine the probability of stopping landing operations at the analyzed airport. The presented methodology for performing analyses can serve to better plan the flight schedule of air carriers, avoiding hours with a high probability of reduced visibility, which may result in the diversion of flights to another airport. 

## 5. The Results of the Assessment of the Impact of the ILS Category on the Performance of Landing at Selected Airports 

### 5.1. Short Description of the Selected Airports

The selected airports for research were Chopin Airport, Warsaw Modlin Airport, and Kraków Airport. Each of these airports has a different category under the ILS, which translates into the fact that landing can be performed under different meteorological conditions (Table 2).

In order to assess the impact of airport navigation equipment on the performance of flight operations, METAR dispatches containing current meteorological data, such as RVR visibility and vertical visibility, among others, were analyzed. An analysis of METAR dispatches totaling approximately 52,000 was prepared to assess the impact of ILS airport navigation equipment on the feasibility of landing at selected airports. METAR dispatches are prepared for an airport, regardless of the type of air carrier. RVR and vertical visibility data were extracted from each dispatch for each day and hour. A quantitative analysis was then conducted to determine the feasibility of landing operations with a particular ILS equipment. Due to unfavorable meteorological conditions, this made it possible to calculate the total time for holding landing operations. Examples of the dispatches are shown in Table 3.

### 5.2. Modlin Airport

Warsaw Modlin Airport has had an ILS cat. II since 2014 (current as of 30 January 2022). This allows landing to be carried out with an RVR of at least 300 m and vertical visibility of not less than 30 m (100 ft).

Figure 4 shows the visibility along the runway (RVR) on selected days in 2019 and 2022 (red line means acceptable minimum visibility RVR, while the blue boxes indicate visibility from METAR). On 23 October 2019, the worst conditions of 2019 prevailed. For comparison, the RVR visibility on 23 October 2022 is indicated. It can be seen from this figure that aircraft landing was halted on this day due to insufficient RVR visibility for 9.5 h. If a category IIIA ILS had been installed on the runway, this time would have dropped to only 2 h. On the other hand, if there had been a category I system, then this time would have increased to 12 h. The longest period in which visibility was below acceptable values lasted from 2:00 a.m. to 8:30 a.m. The lowest visibility was recorded at 2:00 a.m. and 2:30 a.m. and was 150 m. The hours when the visibility along the runway was more than 2000 m and was not operationally significant have been omitted from Figure 5. 

The research shows that landings for 2019 could not be performed for 60.5 h. Figure 5 shows the months in which landings were most frequently halted. This analysis shows that the number of hours when landings could not be performed was highest in October. Reduced visibility mainly occurred during the autumn period. The drop in visibility below acceptable values was sporadic for the other seasons. From March to July, aircraft could perform landing without any interruption. 

Analyzing the years from 2019 to 2022, it appears that in 2019, there were definitely worse visibility conditions than in other years, which is why 2019 was included in the following results for all analyzed airports.

Figure 6 shows the hours when landing operations were most often suspended throughout 2019. From 0:00 to 8:00, landing operations could not be performed for a total of 35.5 h and from 18:00 to 24:00 for 21 h. In the time range of 00–16:00, throughout 2019, there were no problems with limited visibility.

### 5.3. Chopin Airport

Chopin Airport has had a category IIIA ILS on runway 3 in direction 33 since 2018 and a category II system on runway 1 in direction 29 (as of 30 January 2022). Figure 7 shows the visibility along the runway (RVR) on 25 October 2019 when meteorological conditions were among the worst of the year. Landings were not suspended on the aforementioned day. The RVR visibility at 18:30 and 19:00 reached 200 m, which would have led to a halt in the possibility of landing for lower-category ILSs. The hours when the RVR visibility was over 2000 m were omitted as they were not operationally relevant.

The vertical visibility on 25 October 2019 is shown in Figure 8. Landings were not halted once due to poor vertical visibility. Between 19:00 and 20:30, the vertical visibility dropped to 30 m, but this did not result in restrictions on flight operations with the existing ILS cat. IIIA.

Landings were not halted during the time interval analyzed due to low visibility. Continuity in conducting landing was maintained due to the installed ILS category IIIA. In addition, there were relatively good meteorological conditions at EPWA airport.

### 5.4. Kraków Airport 

Kraków Airport has a cat. I ILS (as of 30 January 2022). It can be seen from Figure 9 that the RVR visibility fell below the permissible value, and landings could not take place for 14.5 h in total.

Figure 10 shows the total number of hours without landing operations by month. Landing operations were stopped most frequently during the autumn season. In October, landing procedures were halted for 39 h, while in November, they were halted for as many as 86 h.

Figure 11 shows the hours at which landing operations at EPKK aerodrome were most often suspended in 2019. Figure 11 clearly shows that the greatest problems with visibility occurred between 00:00 and 8:00. Throughout the year, in this period, landing operations, due to limited visibility, were suspended for a total of 112.5 h. The most favorable visibility conditions occurred between 11:00 and 16:00, during which, throughout 2019, landing operations were not suspended even once.

## 6. Results and Discussion

The article discusses the impact of different categories of instrument landing systems (ILSs) on the ability to conduct landing procedures at airports similarly to [51]. It highlights the relationship between ILS categories, meteorological conditions, and the duration of halted landing operations at these airports. The text also provides data from Table 4, which demonstrates how upgrading ILS categories can significantly reduce the probability of halted landings during specific time periods compared with study [52]. It emphasizes the critical role of ILS in ensuring smooth airport operations, particularly in adverse weather conditions following the particular results of [53]. At each analyzed airport, a different category of ILS was installed, which directly impacts the ability to perform landing procedures. 

Based on the data collected in Figure 12, Katowice Airport could not operate aircraft for more than 160 h. This is not only because the lowest category of ILS was installed there but also because the worst meteorological conditions prevail there. Particularly in autumn periods, visibility drops are very common there. At EPMO airport, operations were halted for a total of 60.5 h, while at EPWA airport, they were not halted once. It is worth noting that the lack of halted landing operations at EPWA airport is not only due to the high category of the ILS but also to the relatively favorable meteorological conditions there.

A category II ILS is installed at EPMO airport. If a category III ILS was installed at the airport, this would reduce the number of hours of operational delays to 20.5 h. If a category I ILS was present, this time would increase to as much as 101.5 h. 

At EPWA airport, which has a category IIIA ILS, landings were not halted once in 2019 due to poor RVR and vertical visibility. If a category II ILS had been installed at the airport, operations would have been stopped for 5.5 h, whereas they would have been stopped for 24 h with an ILS I system. The reason for this is the relatively good meteorological conditions.

In 2019, landing could not be carried out at EPKK airport for 160.5 h. The presence of a category II ILS would bring the time down to 80.5 h, while with category IIIA, it would be 60 h. Introducing a category II ILS would significantly increase the ability to perform landings (by 80 h). The higher category (II) would allow a landing procedure with an RVR of not less than 300 m and a decision height of not less than 30 m. A Category II ILS would reduce the number of delays and the number of aircraft diverted to alternate airports. A higher category would also affect the level of safety in performing landings. Introducing category IIIA would not reduce the number of hours during which landing procedures are not possible as significantly as moving from ILS category I to ILS category II. Category I is the lowest category of ILS. Table 4 shows the percentage of time-stopping landings with a category II ILS in November. The data show that there would be a significant decrease in time in each period. It is worth noting that in the 2:00–4:00 period, with category I, the percentage of time-stopping landings was 12.92%, while with category II, it would drop to only 4.58% (a change of 8.35%). An equally spectacular change would occur between 4:00 and 6:00 from 11.25% to 4.17%. Only between 18:00–20:00 would the time-stop remain unchanged at a further 3.33%. From Table 4, it is also possible to compare the time-stop at category IIIA. The operation interruption is identical in all hours except 6:00–8:00. Therefore, the introduction of category IIIA would slightly improve the airport operation in November. Changing the ILS category from I to II would significantly reduce the number of hours when landing procedures cannot be performed.

## 7. Conclusions

The impact of meteorological conditions on the possibility of carrying out flight operations was assessed using METAR dispatches from the whole of 2019 and 2022 for EPWA, EPMO, and EPKK airports. An analysis of meteorological conditions in 2019–2022 showed that the worst conditions in terms of visibility were observed in 2019. For this reason, the analysis was made on the basis of 2019 data. For EPMO airport, meteorological conditions prevented landing operations for a total of 60.5 h in 2019. Landings were stopped most frequently in October and November. At EPWA airport, landing operations were not stopped even once in 2019. At EPKK airport, a landing was stopped for a total of 160.5 h. This occurred most frequently in November and October. The results of the research show at three analyzed airports that the greatest problems with meteorological conditions were at EPKK airport, which has the lowest category of ILS. It also shows for how long in 2019 landing procedures would be halted with other ILS categories (lower and higher). For EPMO airport, the introduction of a higher ILS category IIIA would reduce the number of hours during which landings could not be carried out from 60.5 to 20.5 h. These figures show that introducing a higher category ILS at this airport is needed and would also reduce delays, diversions, and costs. If there had been a category I ILS at this airport, the time during which operations could not be carried out would have increased from 60.5 to 101.5 h, which shows that a category II ILS was rightly introduced and allows for a significant reduction in this figure. It also shows that there are relatively difficult meteorological conditions at the airport. At EPWA airport, where landing procedures were not stopped even once in 2019 due to meteorological conditions, if category II had been installed instead of the ILS IIIA system, the number of hours during which landings could not be carried out would have increased to only 5.5 h, whereas with category I, it would have increased to 24 h. This shows that at EPWA airport, the introduction of category IIIA was not as necessary as at EPMO airport. At EPKK airport, if category II of the ILS had been introduced, this would have reduced the number of hours during which landing procedures could not be performed by half, from 160.5 to 80.5 h. With category IIIA, it would be reduced to 60.5 h. Based on the research results, it is possible to declare that EPKK airport needs a higher category ILS than EPWA and EPMO airports. Moreover, an increase not to II but to IIIA should be considered because the airport is close to rivers and has unfavorable meteorological conditions. 

The article presents a quantitative analysis showing that visibility is important for landing operations conducted with the use of the ILS. The impact of the visibility parameter on the implementation of landing operations at the tested airports was indicated. Among the research works, meteorological parameters, in particular wind and its impact on the implementation of air operations, were analyzed. The adopted assumptions regarding visibility do not include factors that affect visibility, namely the retention of certain air masses, air temperature and humidity, and the local microclimate of the airport surroundings. Forecasting visibility-reducing phenomena is an important task for air traffic control. In modern aviation, the most modern technologies are used to pilot the aircraft in even the most difficult weather conditions. Nevertheless, the visual assessment of the situation in the environment, made by the pilot during take-off, landing, and the flight itself, remains a key element of aviation safety. It is obvious, therefore, that phenomena limiting visibility fundamentally affect the possibility of conducting air traffic. The level of cloud cover and the type of clouds are of great importance for the safety of air traffic. Such studies were conducted in order to check and identify the factors affecting the safety of air traffic. The presented analysis complements the research area concerning the impact of visibility on the implementation of air traffic in the aspect of airport navigation equipment. The method of proceeding in order to determine the probability of the inability to carry out air traffic was indicated. The developed course of action and the established probability of stopping landing operations may be important in the airline’s flight planning recommendations to avoid delays. Further research work may concern the indication of working hours using the GNSS approach. This approach can be valuable from the point of view of safe landing.

## Figures and Tables

**Figure 1 sensors-23-07953-f001:**
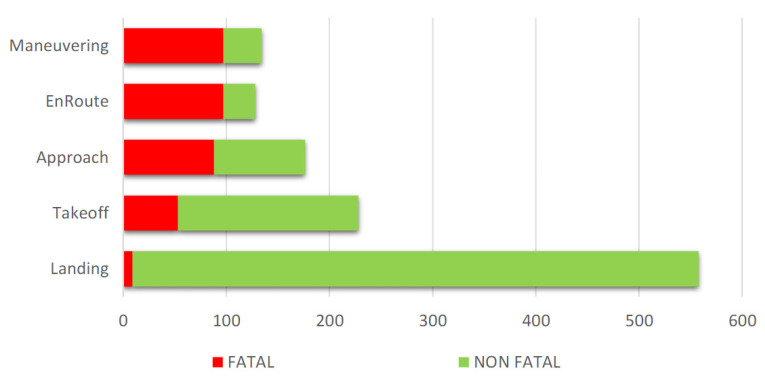
Relation of weather-related accidents to phase of flight operations for years 2009 to 2018 [10].

**Figure 2 sensors-23-07953-f002:**
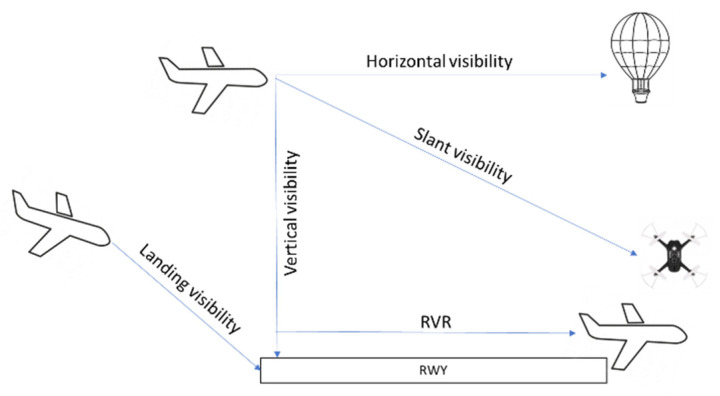
Type of visibility.

**Figure 3 sensors-23-07953-f003:**
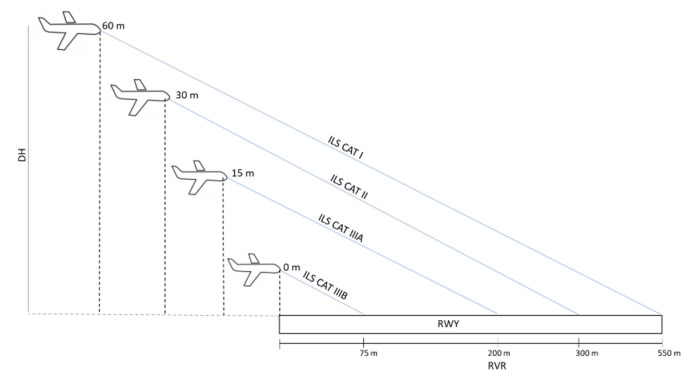
ILS categories. Sources: own elaboration based on [40].

**Figure 4 sensors-23-07953-f004:**
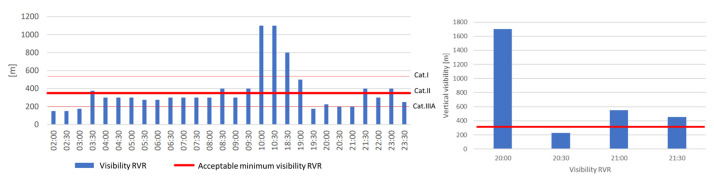
Visibility RVR at Modlin Airport on 23.10.2019 and 23 October 2022.

**Figure 5 sensors-23-07953-f005:**
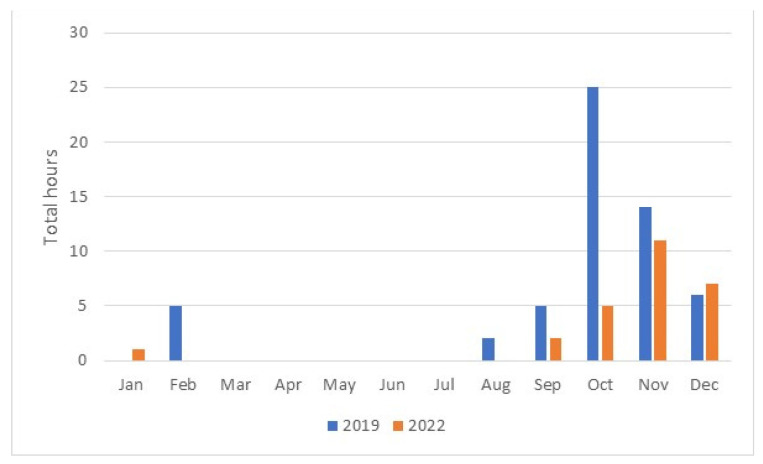
Number of hours per month with reduced flight operations in 2019 and 2022.

**Figure 6 sensors-23-07953-f006:**
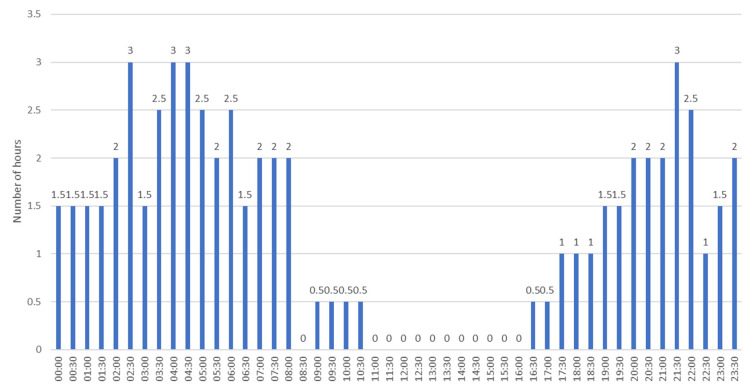
The number of hours when landing operations were most frequently held back at EPMO airport.

**Figure 7 sensors-23-07953-f007:**
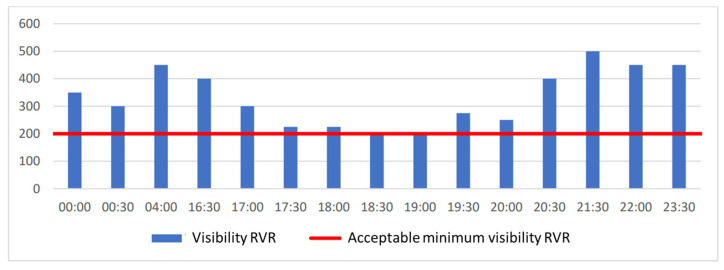
Visibility of RVR on 25 October 2019.

**Figure 8 sensors-23-07953-f008:**
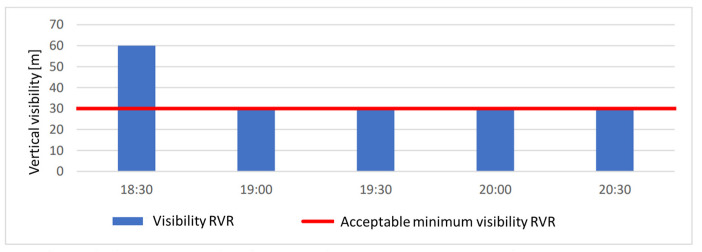
Vertical visibility on 25 October 2019.

**Figure 9 sensors-23-07953-f009:**
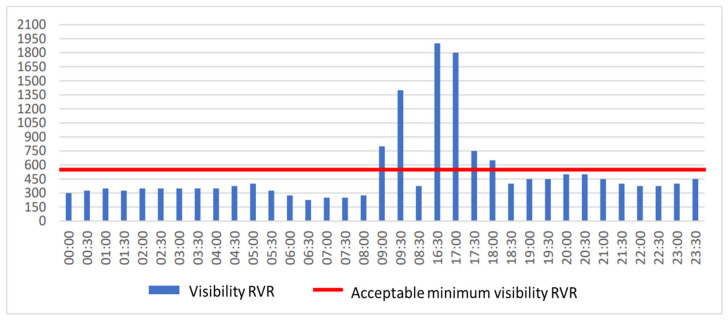
Visibility of RVR on 1 November 2019.

**Figure 10 sensors-23-07953-f010:**
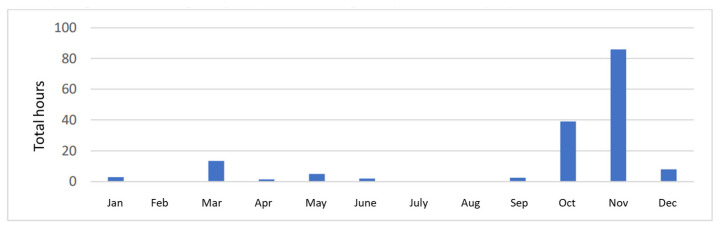
Number of hours per month with reduced flight operations in 2019.

**Figure 11 sensors-23-07953-f011:**
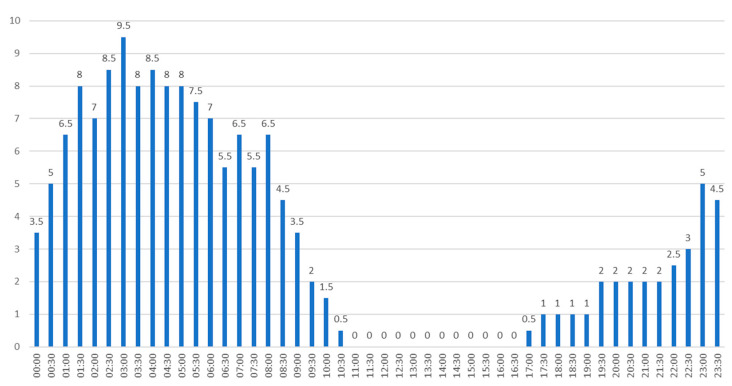
The number of hours when landing operations were most frequently suspended at Krakow Airport.

**Figure 12 sensors-23-07953-f012:**
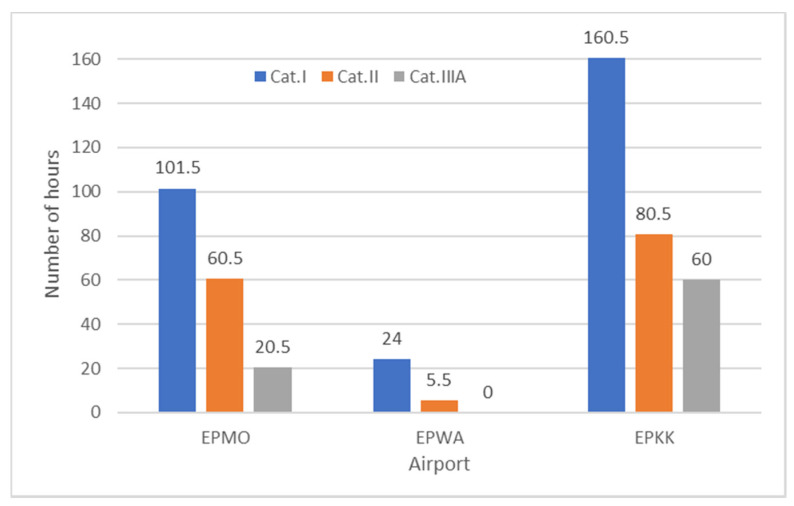
Number of hours with reduced flight operations at selected airports.

**Table 1 sensors-23-07953-t001:** Visibility limitations depend on the phenomena.

The Phenomena	Visibility (km)		
	Poor Intensity	Middle Intensity	Strong Intensity
drizzle	3–2	2–1	<1
light rain	10–6	6–4	<1
rain	5–4	4–2	2–1
snow	>3	3–1.5	1.5–0.5
haze	10–4	4–2	2–1
fog	1–0.5	0.5–0.2	<0.2

Source: [39].

**Table 2 sensors-23-07953-t002:** Categories of ILS and codes of selected airports.

Airport	Category of ILS	Code ICAO
Kraków	I	EPKK
Modlin	II	EPMO
Chopin	IIIA	EPWA

Source: own elaboration (actual on 3 December 2022).

**Table 3 sensors-23-07953-t003:** Example of a set of METAR messages with selected marked RVR and vertical visibility.

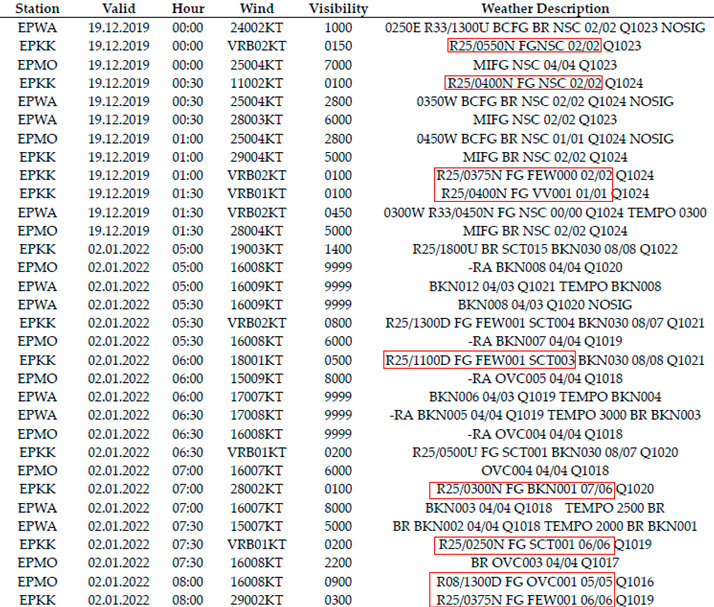

Red box—example description of Runway Visual Range (RVR).

**Table 4 sensors-23-07953-t004:** The time-stop of landing operations at EPKK airport (%).

Hours	Cat. I	Cat. II	Cat. IIIA
0:00–2:00	10.0	6.67	6.67
2:00–4:00	12.92	4.58	4.58
4:00–6:00	11.25	4.17	4.17
6:00–8:00	9.58	5.00	3.33
8:00–10:00	7.92	3.33	3.33
10:00–12:00	1.67	0.42	0.42
12:00–14:00	0.00	0.00	0.00
14:00–16:00	0.00	0.00	0.00
16:00–18:00	1.25	0.83	0.83
18:00–20:00	3.33	3.33	3.33
20:00–22:00	5.00	4.58	4.58
22:00–24:00	8.75	5.83	5.83

## Data Availability

Not applicable.

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
