# Peer review of "The Influence of Visibility on the Opportunity to Perform Flight Operations with Various Categories of the Instrument Landing System"

_sensors, 2023, doi:10.3390/s23187953_

Round 1

Reviewer 1 Report

Review in attachment

Reviewer 2 Report

Recommendation: Major Revision

 Comments:

1.         The exposition on the methodology necessitates a more intricate elaboration, especially concerning the criteria employed in airport selection and the specific repositories of data harnessed. Such elucidation would bolster the study's reproducibility and cogency.

2.         The empirical analysis employed could manifest greater rigor. Incorporating sophisticated statistical modalities, like regression analysis, might illuminate more profoundly the nexus between visibility and landing procedures.

3.         The conclusions drawn from the study demand a more subtle interpretation. Although the results intimate the indispensability of superior category ILS systems for particular airports, it remains imperative to accentuate the study's constraints and the imperative for additional inquiry to corroborate these assertions.

4.         The discourse would be enriched by a more exhaustive exploration of the ramifications emanating from the study's discoveries pertaining to aviation safety and overarching policies. This would adeptly embed the study within the vast tapestry of aviation scholarship.

5.         The narrative would be enhanced by offering more perspicuous elucidations of technical lexicon and theories, especially for those readers whose acquaintance with aviation safety might be cursory.

6.        The article's potency could be augmented by the incorporation of more illustrative tools, such as diagrams or tabulations, thereby vividly delineating the study's conclusions and rendering them more palpable to its audience.

Reviewer 3 Report

This study presents an interesting topic that requires further inquiry. We all agree that the phenomenon is of interest to aviation professionals. However, there are some flaws that need to be addressed before it is considered for publication.

The literature review is too lengthy and lacks studies regarding “aviation and meteorology” and/or “weather related aircraft accidents”. It is suggested authors should briefly explain 1-2 accidents in which the weather / meteorological conditions were a contributing factor. Then, authors should briefly describe findings of previous studies related to aircraft accidents / incidents and weather conditions (visibility?). (e.g., https://link.springer.com/article/10.1007/s00024-019-02168-6 - https://journals.sagepub.com/doi/epdf/10.1177/154193120104500241).

There is some sort of repetition throughout the literature review. For example, authors mentioned / explored “visibility” in line 83 but also in a new section of the manuscript (line 213).

In line 51 authors indicated the “NTSB” had conducted a study […]. However, the author of this study is Long (2022). Yet, Long (2022) only analyzed accidents and incidents involving general aviation aircraft (not explained in this manuscript). 

Authors frequently used the terms “accidents” and “incidents” as if they were the same type of aviation safety events. However, there are key differences between an aircraft accident (e.g., lead to fatalities) and an aircraft incident (e.g., minor damage to aircraft). It is recommended authors should use the ICAO definition of aircraft accident and of aircraft incident to support their claims.

A number of acronyms / abbreviations were used throughout the manuscript without a definition of their meanings (e.g., METAR; ILS; RVR). Readers, especially those without an aviation background will benefit from these definitions / explanations (first time authors use an acronym).

In line 35 authors mentioned “The analysis [6] shows the rate of incidents related to meteorological conditions (Figure 2). However, the title of Figure 2 is “Fatal weather-related accidents (in %)”.

A number of citations is missing (see Lines 165; 181). Some equations may not be necessary for the purpose of this study (see lines 359 & 369).

Some Figures (e.g., Figure 4) in the manuscript are not necessary either.

The manuscript will benefit from a section describing the methodology of the study – what type of data was analyzed? Data analyses processes (e.g., 52,000 METARs)? Period? Validity & reliability? Which type of flights (air carriers? General aviation?) were impacted by weather conditions (RVR)?

The “discussion” should be a new section (line 547).

There are also major flaws in the “discussions and conclusions” sections. For example, authors are expected to compare and contrast their findings with previous research. Any practical applications considering the findings of the study?

Authors should explain the limitations of their study (e.g., a number of factors [unstable approach; technical issues with the aircraft] could lead to a missed approach and / or to a diversion to another airport).

Lastly, a grammar review is needed before this manuscript is accepted for publication.

A grammar review is highly recommended.

Reviewer 4 Report

1. The authors have not given proper affiliation.

2.  The methodology must be very clear. In the present scenario, it is very confusing. 

3. More experimental results are needed, those must be compared with the previous methods and research as well

4.  The authors should incorporate a flow chart of the methodology of the overall study in the introduction section for further clearance of objective

5.  Overall formatting and english and preposition require severe attention. 

Round 2

Reviewer 1 Report

1. I still can't find a description of the test method. In response, the authors wrote that it has been supplemented and marked in blue, but in the attached file, there is no blue font in any fragment. Does this supplement apply to section 3? 

2. I still miss the Discussion section, where I should refer to the results of other authors in the researched area. In my opinion, this section still needs to be completed.

3. Figure 2 does not have a given source and is based on the results of research by other authors, as I understand it.

Author Response

We would like to thank the Reviewers for their time and effort to read, evaluate and comment on our paper. Please find a detailed discussion of the changes we have introduced to the paper.

  1. I still can't find a description of the test method. In response, the authors wrote that it has been supplemented and marked in blue, but in the attached file, there is no blue font in any fragment. Does this supplement apply to section 3? 

Response 1: The paper has been rebuilt and chapter 3 has been added, in which we have described the data and the method of conducting analyses.

  1. I still miss the Discussion section, where I should refer to the results of other authors in the researched area. In my opinion, this section still needs to be completed.

Response 2: During the revision process of the article, we have been thinking a lot about interpretation of literature review and discussion, so we completely rewrite the introduction and we added the references on next previous studied scientific papers to introduction and revised and completed chapter concerning results and discussion, where are refer the results of other authors.

  1. Figure 2 does not have a given source and is based on the results of research by other authors, as I understand it.

Response 3: After rewriting of the introduction we only interpreted this previous figure 2 and added the reference.

Reviewer 2 Report

Accept.

Author Response

Thank you very much.

Reviewer 3 Report

As previously mentioned, the literature review is too lengthy and lacks studies regarding “aviation and meteorology” and/or “weather related aircraft accidents”. It is suggested authors should briefly explain 1-2 accidents in which the weather / meteorological conditions were a contributing factor. Then, authors should briefly describe findings of previous studies related to aircraft accidents / incidents and weather conditions (visibility?).

There is some sort of repetition throughout the literature review. For example, authors mentioned / explored “visibility” in line 83 but also in a new section of the manuscript (line 213).

In line 65 authors indicated the “NTSB” had conducted a study […]. However, the author of this study is Long (2022). Yet, Long (2022) only analyzed accidents and incidents involving general aviation aircraft (not explained in this manuscript). 

In line 54 authors presented the ICAO definitions of "aviation" accident and incident. However, the ICAO Annex 13 provides definitions of "aircraft" accident and incident. Yet, authors defined accident as an incident.

A number of acronyms / abbreviations were used throughout the manuscript without a definition of their meanings (e.g., METAR; ILS; RVR). Readers, especially those without an aviation background will benefit from these definitions / explanations (first time authors use an acronym).

A number of citations is still missing (see lines 54 through 64).

There are also major flaws in the “discussions and conclusions” sections. For example, authors are expected to compare and contrast their findings with previous research. Any practical applications considering the findings of the study?

Authors should explain the limitations of their study (e.g., a number of factors [unstable approach; technical issues with the aircraft] could lead to a missed approach and / or to a diversion to another airport).

Lastly, a grammar review is needed before this manuscript is accepted for publication.

N/A

Author Response

We would like to thank the Reviewers for their time and effort to read, evaluate and comment on our paper. Please find a detailed discussion of the changes we have introduced to the paper.

  1. As previously mentioned, the literature review is too lengthy and lacks studies regarding “aviation and meteorology” and/or “weather related aircraft accidents”. It is suggested authors should briefly explain 1-2 accidents in which the weather / meteorological conditions were a contributing factor. Then, authors should briefly describe findings of previous studies related to aircraft accidents / incidents and weather conditions (visibility?).

Response 1: The literature chapter has been rebuilt. We believe that providing many examples will be valuable to the reader because this is an important area.

  1. There is some sort of repetition throughout the literature review. For example, authors mentioned / explored “visibility” in line 83 but also in a new section of the manuscript (line 213).

Response 2: The paper has been rebuilt.

  1. In line 65 authors indicated the “NTSB” had conducted a study […]. However, the author of this study is Long (2022). Yet, Long (2022) only analyzed accidents and incidents involving general aviation aircraft (not explained in this manuscript). 

Response 3: This part has been rebuilt.

  1. In line 54 authors presented the ICAO definitions of "aviation" accident and incident. However, the ICAO Annex 13 provides definitions of "aircraft" accident and incident. Yet, authors defined accident as an incident.

Response 4: The part has been rebuilt as the additional subsection.

  1. A number of acronyms / abbreviations were used throughout the manuscript without a definition of their meanings (e.g., METAR; ILS; RVR). Readers, especially those without an aviation background will benefit from these definitions / explanations (first time authors use an acronym).

Response 5: We have added appropriate explanations to all abbreviations in the article.

  1. A number of citations is still missing (see lines 54 through 64).

Response 6: We tried to add missing reference.

  1. There are also major flaws in the “discussions and conclusions” sections. For example, authors are expected to compare and contrast their findings with previous research. Any practical applications considering the findings of the study?

Response 7: During the revision process of the article, we have been thinking a lot about interpretation of literature review and discussion, so we completely rewrite the introduction and we added the references on next previous studied scientific papers to introduction and revised and completed chapter concerning results and discussion, where are refer the results of other authors.

  1. Authors should explain the limitations of their study (e.g., a number of factors [unstable approach; technical issues with the aircraft] could lead to a missed approach and / or to a diversion to another airport).

Response 8: We tried to explain in a few words.

  1. Lastly, a grammar review is needed before this manuscript is accepted for publication.

Response 9: A language review and revision of the paper was made again.

Reviewer 4 Report

thank you for the revisions

Author Response

Thank you very much.